# Analysis of the Effects of *Beauveria bassiana* Appressorium Formation on Insect Cuticle Metabolism Based on LC-MS

**DOI:** 10.3390/jof11080595

**Published:** 2025-08-15

**Authors:** Jiarui Chen, Wenzhe Li, Canxia Wu, Songqing Wu, Yinghua Tong

**Affiliations:** Forestry College, Fujian Agriculture and Forestry University, Fuzhou 350002, China; a568484991@163.com (J.C.); liwz20000423@163.com (W.L.); ktaehyun698@gmail.com (C.W.); dabinyang@126.com (S.W.)

**Keywords:** *Beauveria bassiana*, appressorium, pathogenicity, insect epidermis, metabolism

## Abstract

The appressorium is a specialised infection structure formed by *Beauveria bassiana* during host invasion. This study used sulforaphane to regulate the formation rate of *B. bassiana appressoria*, evaluated the correlation between appressorium formation and fungal pathogenicity, and explored its impact on insect cuticular metabolism. The results showed that sulforaphane significantly modulated appressorium formation. Spore suspensions with varying appressorium formation rates were injected into *Opisina arenosella* and *Bombyx mori* larvae. As the appressorium formation rate increased, *B. bassiana* exhibited enhanced pathogenicity, leading to accelerated larval mortality. A significant positive correlation (*p* ≤ 0.05) was observed between appressorium formation and pathogenicity. LC-MS analysis revealed that, prior to appressorium development, larvae activated defence mechanisms involving secondary metabolites, hormone signalling, and toxin metabolism pathways. Following appressorium formation, 61 unique cuticular compounds were identified, along with activation of host lipid metabolism (notably glycerophospholipid degradation), programmed cell death pathways (ferroptosis, necroptosis), and enhanced energy metabolism via the citric acid cycle—collectively indicating disruption of the epidermal defence barrier. Overall, appressorium development by *B. bassiana* significantly reshapes the metabolic landscape of the larval cuticle, thereby enhancing fungal virulence. This study provides a theoretical foundation for understanding the pathogenic mechanisms of *B. bassiana*.

## 1. Introduction

Hypocreomycetidae, of the order Hypocreales and the family Cordycipitaceae. *B. bassiana* has a broad host range, capable of infecting more than 700 insect species across 15 orders and 149 families, as well as over 10 species of ticks and mites from 6 families [1]. It was widely used in the biological control of agricultural and forestry pests, including *Dendrolimus*, *Hyphantria cunea*, *Monochamus alternatus*, *Basilepta melanopus*, and *Ostrinia furnacalis* [2,3,4,5,6].

The infection process of *B. bassiana* is complex. Its conidia adhere to the insect cuticle and germinate into germ tubes. The apex of the germ tube differentiates into infection structures such as appressoria and penetration pegs. Subsequently, the fungus penetrates the insect cuticle through the synergistic action of mechanical pressure and hydrolytic enzymes [7]. Among these structures, appressorium formation is a prerequisite for the successful colonization of the host integument [8]. Feifei Luo et al. have reported that, during appressorium formation, *B. bassiana* produces high levels of acylcarnitines and phospholipids, which help maintain turgor pressure and supply the energy required for host penetration [9]. The appressoria exhibit intense metabolic activity, producing proteases, chitinases, lipases, and other enzymes capable of degrading the insect cuticle. Meanwhile, mechanical pressure generated by the penetration peg assists in breaching the host integument [8].

Previous studies, both domestic and international, have primarily focused on the influence of physical and chemical factors on appressorium formation, as well as its role during host invasion [10,11,12,13]. However, the specific chemical cues on the host cuticle that trigger or modulate this process remain poorly characterized, representing a significant knowledge gap in understanding the fungus–host interaction. We hypothesized that the epidermal chemical composition of the insect cuticle directly influences the rate of *B. bassiana* appressorium formation and, consequently, its pathogenic success. To address this, this study aims to (1) investigate the relationship between appressorium formation rate and fungal pathogenicity, and (2) identify key changes in the epidermal chemical profile of insect cuticles associated with the appressorium formation stage. By employing LC-MS-based non-targeted metabolomics, we seek to provide a theoretical foundation for elucidating the pathogenic mechanisms of *B. bassiana* and its potential applications in biocontrol strategies.

## 2. Materials and Methods

### 2.1. Test Strains and Insects

The test strain was *B. bassiana* bcm-01, which was preserved by the Forest Protection Teaching and Research Section of Fujian Agriculture and Forestry University. The strain was rejuvenated with *O. arenosella* larvae three times, inoculated on PDA plate culture medium, and cultured at 26 °C. The test insects were *O. arenosella* larvae (collected from 26.969° N, 118.972° E, Xiang’an District, Xiamen City, Fujian Province) and *B. mori* larvae (laboratory-reared from an in-house incubated).

### 2.2. Regulation of the Proportion of Appressorium Formation in Beauveria bassiana

Different concentrations of sulforaphane (Glpbio, Montclair, CA, USA) were used to regulate the appressorium formation rate of *B. bassiana* [14]. Sulforaphane (purity: 98.00%) was added to a sterile aqueous solution containing 1 mg/mL of yeast extract (Coolaber, Beijing, China) and trehalose (Glpbio, USA) to prepare five working solutions with sulforaphane concentrations of 0, 0.01, 0.02, 0.05, and 0.10 mg/mL, designated as SF0, SF0.01, SF0.02, SF0.05, and SF0.10, respectively. Spore powder of *B. bassiana* was added to each solution to obtain spore suspensions at a final concentration of 5 × 10^7^ spores/mL.

The suspensions were incubated in a shaker at 26 ± 1 °C and 150 rpm. At 12 h, 24 h, 36 h, 48 h, and 72 h post-inoculation, 100 μL of each suspension was sampled for microscopic observation. For each time point, five fields of view were randomly selected, and the number of germinated spores and those forming appressoria was recorded. Each sulforaphane concentration represented a treatment group, and five replicates were performed per group.

### 2.3. Determination of the Pathogenicity of Different Appressorium Formation Ratios of Beauveria bassiana to Insects

Inoculation was performed using the immersion method [15]. The same five mixed solutions (SF0, SF0.01, SF0.02, SF0.05, SF0.10) as those used in Section 2.2 were used to prepare a spore suspension of *B. bassiana* at a concentration of 1 × 10^8^ spores/mL. Third-instar *O. arenosella* larvae and *B. mori* larvae were immersed in the suspension, with their heads exposed to the liquid surface. After 15 s, they were gently blotted dry on filter paper. They were then placed in an artificial climate chamber at a temperature of 26 °C and a light intensity of L: D = 14:10. Larvae of *O. arenosella* and *B. mori* were treated at five different appressorium formation ratios, with five biological replicates per treatment group and twenty test insects per replicate. Larvae immersed in sterile water were used as the control (CK1 group), and larvae immersed in a sterile solution containing yeast extract (1 mg/mL), trehalose (1 mg/mL), and sulforaphane (0.1 mg/mL) were used as the control (CK2 group) for non-toxicity verification. After inoculation, the mortality of larvae in each treatment was regularly observed and recorded daily, and the observations and statistics were continuously monitored for 7 d.

Similarly, five test worms were inoculated for each treatment. After 48 h, adhesive conidium formation on the worm body was observed under a fluorescence microscope (Nikon, Tokyo, Japan). From the mid-region of the larval prothorax and thoracic dorsum, a 5 mm^2^ section of the dorsal epidermis was excised using sterile surgical scissors. Excess tissue and fat on the epidermis’ inner side were then scraped off and spread flat on a sterile glass slide [16]. A 10 μL LB solution (Shanghai Macklin Biochemical Co., Ltd., Shanghai, China) (0.001% Calcofluor White M2R staining solution) and a 15 μL A solution (containing 10% KOH and 10% glycerol aqueous solution (Xilong Scientific Co., Ltd., Shantou, China) were sequentially applied. After staining for 60 s, the spores were rinsed with distilled water for 3 s, and the number of spores that germinated and formed appressorium was observed under a microscope and counted [17,18]. Five fields of view were randomly examined for each sample, and 20 spores were counted per field.

### 2.4. Effects of Beauveria bassiana Appressorium Formation on Insect Cuticle Metabolism

#### 2.4.1. Appressorium Formation Rate of *Beauveria bassiana* on Larval Cuticles

The fourth instar larvae of *O. arenosella* were inoculated with a spore suspension of *B. bassiana* at a concentration of 1 × 10^8^ spores/mL by the immersion method. The rearing and observation methods after inoculation were the same as those described in Section 2.3. Starting from 12 h after inoculation, five *O. arenosella* were randomly selected every 4 h, and the dynamics of appressorium formation were observed under a fluorescence microscope. The observation protocol followed the same methodology as described in Section 2.3, each sulforaphane concentration was treated as a separate treatment group, with 5 replicates per treatment.

#### 2.4.2. Sample Processing

The inoculation concentration and method are the same as those in Section 2.4.1. Before and after the formation of the appressorium, we randomly selected the larvae of ten *O. arenosella* and anesthetized them on ice until immobilized. The head and terminal abdominal segments were removed using sterile scissors. The body was then dissected longitudinally along the lateral side, and a 5 mm^2^ section of the mid-region of the dorsal epidermis of the prothorax was carefully excised for analysis. Excess internal tissues and fat were gently removed, and the fresh epidermal layer was scraped for further analysis [16]. A control group (ck) of *O. arenosella* larvae, immersed in sterile water, was established and processed concurrently during the same time periods. The sample was then placed in a 2 mL sterile cryotube and frozen in liquid nitrogen for 1 min, and this process was repeated 5 times. We took 100 mg of the epidermis tissue after grinding with liquid nitrogen, placed it in an EP tube, and added 500 μL of methanol aqueous solution (80.00%). This was then vortexed and shaken, placed in an ice bath for 5 min, and centrifuged at 15,000× *g* for 20 min at 4 °C. We took the supernatant, diluted it with mass spectrometry grade water to a methanol content of 53%, centrifuged it for 20 min, and injected it into LC-MS for analysis [19].

#### 2.4.3. LC-MS Analysis

The sample obtained in 1.4.2 was chromatographed using a Vanquish ultra-high performance liquid chromatograph (Thermo Fisher Scientific, Waltham, MA, USA) with a Waters ACQUITY UPLC BEH Amide (Thermo Fisher Scientific, USA) (2.1 mm × 100 mm, 1.7 μm) liquid chromatography column. The liquid chromatography phase A was the aqueous phase, the ammonium acetate concentration was 25 mmol/L, the ammonia concentration was 25 mmol/L, and the phase B was acetonitrile. The sample tray was set to a temperature of 4 °C and an injection volume of 2 μL, the sample was injected for each of five biological replicates. Mass spectrometry conditions were as follows: Scan range was selected as *m*/*z* 100–1500. ESI source settings were as follows: Spray voltage: 3.5 kV; sheath gas flow rate: 35 psi; aux gas flow rate: 10 L/min; capillary temp: 320 °C; S-lens RF level: 60; aux gas heater temp: 350 °C; polarity: positive, negative; MS/MS secondary scans were data-dependent scans.

### 2.5. Data Processing

The experimental data were organized using Microsoft Excel 2020 (Microsoft Corporation, Redmond, WA, USA) to calculate means, standard errors, and cumulative mortality. Statistical analyses were performed using IBM SPSS Statistics v25.0 (IBM Corporation, Armonk, NY, USA). Duncan’s new multiple range test was employed for post-hoc multiple comparisons. The original data of insect cuticle compounds were processed by format conversion and material annotation, and then analysed by orthogonal partial least squares discriminant analysis (OPLS-DA) [20], volcano plot analysis and KEGG enrichment analysis [21,22,23,24,25].

## 3. Results

### 3.1. Germination and Appressorium Formation Rates of B. bassiana at Different Sulforaphane Concentrations

The timing and rate of *B. bassiana* appressorium formation under different sulforaphane concentrations were statistically analysed (Appendix A). As shown in Appendix A, during liquid fermentation, the germination rate of *B. bassiana* conidia decreased with increasing sulforaphane concentration, demonstrating a significant dose-dependent effect (*p* < 0.05). By 72 h, the germination rates in all treatment groups had reached 80–90%. Appressorium formation began at 24 h post-germination and stabilized after 48 h. Sulforaphane concentration had a significant effect on the appressorium formation rate. As sulforaphane concentration increased, the rate of appressorium formation declined significantly. The highest formation rate was observed in the treatment without sulforaphane, reaching 64.20 ± 5.67%, indicating that sulforaphane could significantly inhibit the formation of appressorium without significantly affecting the spore germination of *B. bassiana*.

### 3.2. Correlation Analysis Between Beauveria bassiana Appressorium Formation Rate and Pathogenicity

After inoculation with *B. bassiana* spore suspensions exhibiting five different appressorium formation ratios, the relationship between appressorium formation rate and pathogenicity to the two larval species was statistically analysed (Appendix A). As shown in Appendix A, increasing appressorium formation rates were associated with enhanced pathogenicity. At 5 d, 7 d, and 10 d post-inoculation, significant differences in larval mortality were observed among the different appressorium formation treatments for both *O. arenosella* and *B. mori*.

By day 10 post-inoculation, the treatment with the highest appressorium formation rate resulted in corrected mortality rates of 95.50% and 97.89% for *O. arenosella* and *B. mori*, respectively, with corresponding cadaver rates of 89.00% and 97.00%. These values were significantly higher than those of the other treatments. In addition, the lethal time to 50% mortality (LT_50_) was shortened to 5.4 d and 5.8 d for *O. arenosella* and *B. mori*, respectively, indicating a faster killing rate.

No significant difference in mortality was observed between larvae treated with yeast extract, trehalose, and sulforaphane solution and those treated with sterile water, indicating that the formulation components were non-toxic to both larval species. Overall, the appressorium formation rate of *B. bassiana* was positively correlated with its pathogenicity.

### 3.3. Dynamics of Appressorium Formation by B. bassiana on the Cuticle of O. arenosella

The temporal dynamics of appressorium formation by *B. bassiana* on the integument of *O. arenosella* larvae were quantified, and the results are presented in Figure 1. As shown in Figure 1, appressorium appeared on the integument of the larvae of *O. arenosella* 20–24 h after inoculation, and then increased rapidly. After 48 h, the appressorium formation rate reached more than 60.00%, and then stabilized.

### 3.4. Effects of B. bassiana Infection on Differences in the Cuticle Compounds of the Larvae of the O. arenosella

#### 3.4.1. Orthogonal Partial Least Squares Discriminant Analysis (OPLS-DA)

Figure 2 presents OPLS-DA score plots comparing the metabolic profiles of larval epidermis between *B. bassiana*-infected groups (undergoing appressorium formation) and those immersed in sterile water (control groups) at matched time points. As illustrated, the metabolic profiles of treated samples cluster distinctly from those of controls, indicating significant reprogramming of epidermal chemistry during *B. bassiana* infection. The OPLS-DA model demonstrated good predictive ability (Q^2^ > 0.5), and its statistical reliability was confirmed by cross-validated ANOVA (*p* < 0.05). These results demonstrate that *B. bassiana* appressorium formation triggers profound reorganization of host cuticular metabolites. The validated model provides a foundation for identifying differentially abundant metabolites linked to infection mechanisms; thus, further screening of these metabolites can be performed based on this model.

#### 3.4.2. Volcano Plot Analysis

The overall distribution of cuticular metabolites in *O. arenosella* larvae before and after *B. bassiana* appressorium formation was analysed using volcano plots, in comparison with non-inoculated controls at the corresponding time points (Figure 3). As shown in Figure 3, prior to appressorium formation (A), 245 differential cuticular metabolites were identified between the treatment and control groups, including 171 in positive ion mode and 74 in negative ion mode. Among these, 22 metabolites were significantly upregulated and 223 were significantly downregulated.

Following appressorium formation (B), 75 differential metabolites were detected (47 in positive ion mode and 28 in negative ion mode), of which 34 were significantly upregulated and 41 were significantly downregulated. These results indicate that *B. bassiana* infection induces substantial metabolic changes in the larval cuticle, both before and after appressorium formation, suggesting that the fungus extensively interferes with the cuticular metabolic pathways of *O. arenosella*.

#### 3.4.3. Differential Compounds Unique to the Larval Epidermis After the Formation of *B. bassiana* Appressorium

A Venn diagram was generated to compare the differential cuticular metabolites in *O. arenosella* larvae before and after the formation of *B. bassiana* appressoria, relative to the non-inoculated control at the corresponding time points (Figure 4). As shown in Figure 4, 245 differential metabolites were identified before appressorium formation, and 75 were detected afterward. Among these, 231 metabolites were unique to the pre-formation stage, and 61 were unique to the post-formation stage. Fourteen metabolites were shared between both stages.

The unique upregulated metabolites were primarily classified into three categories: organic acids and their derivatives, benzenoids, and amino acids and their metabolites. The most abundant upregulated group was organic acids and their derivatives, including thiamine monophosphate, ethylenediaminetetraacetic acid (EDTA), 3-(3-methoxyphenyl) propionic acid, and 5-hydroxyindole-3-acetic acid, among seven compounds in total. The second major class was benzenoids, with five upregulated compounds, such as 5,5′-dehydrodivanillic acid and oxprenolol. The third class was amino acid-related metabolites, including valyl-aspartic acid and valyl-cystine.

In contrast, the uniquely downregulated metabolites were dominated by amino acid derivatives and benzenoids. Representative downregulated amino acid metabolites included glycyl-tryptophan, Gly-Met-OH, and cysteinyl-tryptophan. Downregulated benzenoids included enilconazole, 4-methylhippuric acid, and harpagoside, among others.

#### 3.4.4. KEGG Enrichment Analysis

As shown in Figure 5, differential cuticular metabolites identified in *O. arenosella* larvae before and after *B. bassiana* appressoria formation (A), compared with corresponding controls, were enriched in 20 metabolic pathways (KEGG, *p* < 0.05, enrichment score > 1.5). Eighteen pathways were uniquely enriched prior to appressoria formation, primarily involving early host immune responses (e.g., hormone receptor signalling), toxin biosynthesis (e.g., aflatoxin production), defensive secondary metabolite synthesis (e.g., flavonoids), and stress responses (e.g., cortisol regulation). With regard to post-appressoria formation (B), 18 distinct pathways were enriched, predominantly related to host energy metabolism (carbohydrates, lipids), programmed cell death (necroptosis, ferroptosis), and fungal nutrient acquisition (steroid degradation). Two pathways—diterpenoid biosynthesis and biosynthesis of secondary metabolites—were enriched at both stages. These results suggest that, before appressoria formation, *O. arenosella* larvae primarily activate immune and stress-related pathways in response to *B. bassiana*. After appressoria formation, the response shifts to energy metabolism and tissue degradation, reflecting a breakdown of metabolic homeostasis in infected larvae.

## 4. Discussion

In our study, we demonstrated a significant positive correlation between the appressorium formation rate of *B. bassiana* and its pathogenicity, with higher formation rates enhancing virulence and accelerating host mortality (*p* < 0.05). Appressoria, as specialized infection structures, facilitate cuticle penetration by forming infection pegs or invasive hyphae and secreting extracellular enzymes such as proteases, chitinases, and lipases that degrade the insect integument [26,27]. Yulian He et al. have found that treatment with tricyclazole, which disrupts turgor pressure during both the development and maturation of appressoria in *Exserohilum turcicum*, significantly reduces the infection efficiency of its host [28]. Based on our observations, we found that sulforaphane-induced appressorial inhibition significantly reduces mortality rates in both *O. arenosella* (95.50–52.80%) and *B. mori* (97.89–61.05%) (Appendix A). This aligns with genetic evidence: MPL1 phosphorylation-site mutants in *Metarhizium anisopliae* exhibit defective appressoria and attenuated virulence [29], while *Magnaporthe oryzae* MoWHI2/MoPSR1 knockouts show impaired host colonization due to appressorial defects [30]. Critically, we extend this paradigm by demonstrating that host cuticular metabolites dynamically regulate appressorium development—a previously underexplored facet of fungal pathogenesis.

In this study, LC-MS-based metabolomic analysis (fold change > 2, *p* < 0.05) identified 61 unique differential cuticular metabolites in *O. arenosella* larvae following *B. bassiana* appressorium formation. These metabolites included organic acids and derivatives, benzenoids, and amino acid-related compounds. These shifts suggest active host–pathogen chemical interplay, with three metabolites warranting detailed discussion. Firstly, there is glycylmethionine, a dipeptide formed by a peptide bond linking glycine and methionine and which functions as a methionine derivative. Beyond its involvement in glutathione synthesis (contributing to antioxidant defence) [31,32], this compound likely modulates fungal virulence through the following mechanisms: (1) Competitive inhibition of methionine transport, where, by interfering with *B. bassiana* methionine transporter functionality, glycylmethionine disrupts the sulphur assimilation pathway, with this impairment compromising the fungus’s capacity to counter host-derived oxidative stress [[31],[32],,[33]]; and (2) inhibition of extracellular protease activity, as dipeptides competitively bind subtilisin-like Pr1 proteases critical for cuticle degradation [34]. Supporting this, *M. anisopliae* Pr1-null mutants show 20% reduced appressorial penetration [35]. Secondly, there is thiamine pyrophosphate (TPP), a key coenzyme in carbohydrate and amino acid metabolism. As a cofactor for pyruvate dehydrogenase (PDH) and transketolase [36,37], reduced TPP likely results in the following: (1) Starvation of appressorial acetyl-CoA: This limits lipid biosynthesis required for turgor pressure generation [38], with an analogous mechanism whereby tricyclazole inhibits melanin-linked turgor in *Exserohilum turcicum*, significantly reducing the infection efficiency of its host [28]; (2) impairment of the pentose phosphate pathway flux: During cuticular invasion, diminished NADPH oxidase activity compromises penetration peg efficiency through the cuticle [39]. Finally, there is 1,2-dioleoyl-sn-glycero-3-phosphocholine (DOPC)—a glycerophospholipid used to model membrane systems—which may accumulate as a host immune strategy. Membrane remodelling could hinder appressorial adhesion or infection peg formation, as glycerophospholipids alter surface hydrophobicity [40,41].

Integration with host immune response, where the metabolite shifts we observed reflect a tripartite interaction: (1) fungal enzyme secretion degrades cuticular components; (2) host countermeasures alter metabolite availability to suppress appressoria; and (3) *B. bassiana* adapts by scavenging nutrients (e.g., sulphur from glycylmethionine) or by exploiting signalling lipids. This finding aligns with previous reports that *Sarcophaga carnaria* L. enhances resistance to *B. bassiana* through cuticular fatty acid remodelling [19].

## 5. Conclusions

Overall, these findings demonstrate that (1) insect cuticles function not as passive barriers but dynamically counteract *B. bassiana* through metabolomic reprogramming; (2) the interaction between *B. bassiana* and the host insect cuticle involves not only mechanical and enzymatic penetration but also complex chemical signalling and metabolic interference. Further studies should elucidate the molecular mechanisms by which cuticular metabolites modulate appressorium development and fungal pathogenicity. Moreover, this work establishes novel conceptual frameworks for investigating fungus–insect cuticle interactions while providing a theoretical foundation for optimizing *B. bassiana* biocontrol applications and deciphering its pathogenesis mechanisms.

## Figures and Tables

**Figure 1 jof-11-00595-f001:**
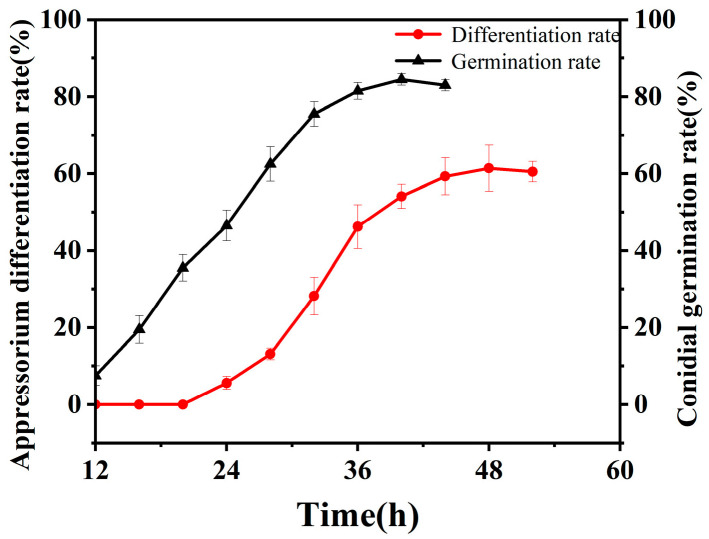
Dynamic changes in the germination rate and attachment cell formation rate of *B. bassiana* conidia on the integument of the larvae of *O. arenosella*.

**Figure 2 jof-11-00595-f002:**
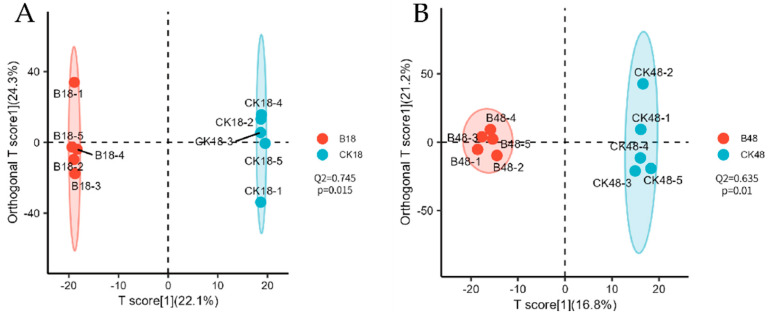
OPLS-DA score plot of insect cuticular compounds before and after appressorial formation by *B. bassiana* compared with time-matched controls. (**A**) Prior to appressorium formation and (**B**) following appressorium formation. Note: The abscissa (*x*-axis) represents predictive component scores, where separation along this axis reflects inter-group variation. The ordinate (*y*-axis) displays orthogonal component scores, with dispersion along this axis indicating intra-group variation. Percentages adjacent to axes denote the proportion of dataset variance explained by each component. Model predictive capability is quantified by Q^2^ values, where Q^2^ > 0.5 indicates a statistically valid model.

**Figure 3 jof-11-00595-f003:**
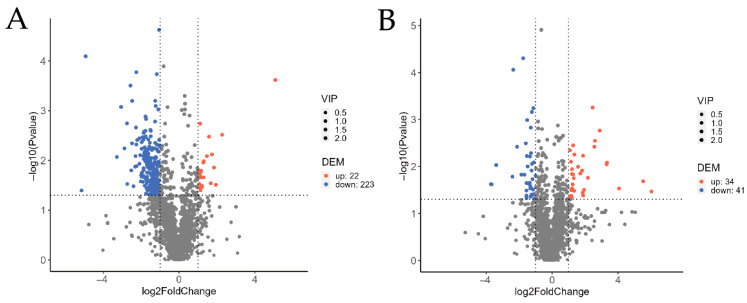
Volcano plot of insect cuticular compounds before and after appressorial formation by *B. bassiana* compared with time-matched controls. (**A**) Prior to appressorium formation and (**B**) following appressorium formation. Note: In the volcano plot, each point represents a metabolite. Significantly upregulated metabolites are denoted by red symbols, while significantly downregulated metabolites are indicated by blue symbols. Symbol size scales with the variable importance in projection (VIP) score. The abscissa (*x*-axis) displays log_2_-transformed fold-change values, where larger absolute values indicate greater differential abundance between experimental groups. The ordinate (*y*-axis) shows −log_10_(*p*-value), with higher values reflecting greater statistical significance of differential expression.

**Figure 4 jof-11-00595-f004:**
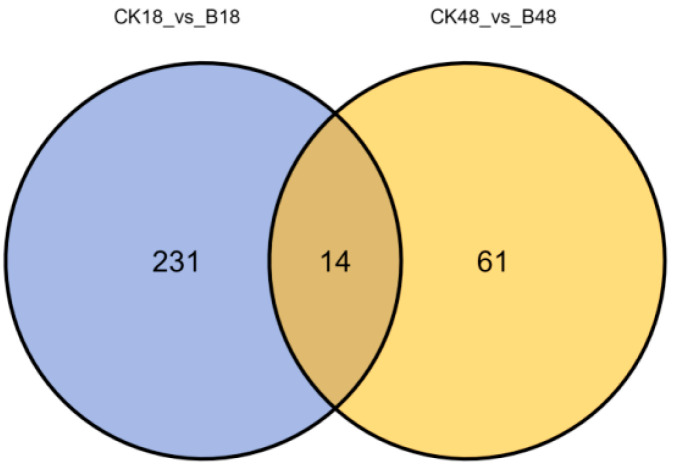
Venn diagram of differential cuticular compounds in insect hosts before and after appressorial formation by *B. bassiana* compared with time-matched controls. Note: B18, inoculated group before appressorial formation (18 h); CK18, control group before appressorial formation (18 h); B48, inoculated group after appressorial formation (48 h); CK48, control group after appressorial formation (48 h).

**Figure 5 jof-11-00595-f005:**
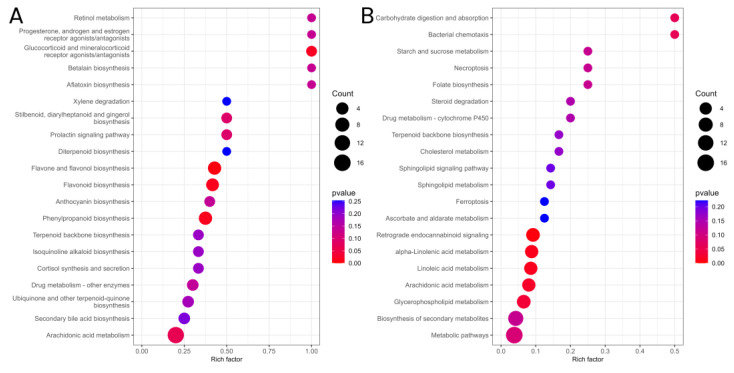
Enrichment analysis of differential cuticular compounds in insect hosts before and after appressorial formation by *B. bassiana* compared with time-matched controls. (**A**) Prior to appressorium formation and (**B**) following appressorium formation.

## Data Availability

The original contributions presented in this study are included in the article/Appendix A. Further inquiries can be directed to the corresponding author.

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
