# Peer review of "Analysis of the Effects of Beauveria bassiana Appressorium Formation on Insect Cuticle Metabolism Based on LC-MS"

_jof, 2025, doi:10.3390/jof11080595_

Round 1

Reviewer 1 Report

The manuscript submitted for publication in the Journal of Fungi is methodologically well designed and thematically relevant, with the potential to contribute to the understanding of how the rate of Beauveria bassiana appressorium formation affects pathogenicity and cuticular metabolism in two insect hosts. The particular value of the work is reflected in the combination of experimental regulation of appressoria via sulforaphane and non-targeted metabolomics (LC-MS), getting new insights into fungal-insect interactions. The work is relevant, and methodologically good, and addresses a key knowledge gap in the biocontrol and fungal pathogenesis field. However, the manuscript has several shortcomings that need to be addressed. Major revision of the paper before publishing would be beneficial. Language and text style (adequate spacing between the words) should be improved.

The paper can be considered for acceptance after major revision.

Below are my specific comments and questions about the manuscript:

Abstract:

Lines 12-13: Please indicate in abstract that sulforaphane was used as a chemical regulator of appressorium formation.

Introduction:

Lines 53-60: These sentences describe the materials and methods used in this experiment, which do not need to be described in the Introduction section. I recommend stating the hypothesis and knowledge gaps, as well as concluding this section with the aims of the study.

Materials and methods:

Line 67: Please indicate the origin of B. mori larvae.

Line 72-73: To make the manuscript easier to read and follow, an abbreviated group name should be formed for each concentration used.

Line 91: In theses sentence you wrote “Larvae immersed in sterile water were used as the control ck1”. I suggest you to write behind the word control (CK1 group).

Line 94: The same as in previous comment.

Line 102: Avoid starting the sentence with number.

Line 105: Here you wrote 1 minute, at other places you wrote s for seconds and d for days. Decide weather you will use abbreviation or full words throughout the hole manuscript.

Line 117: “Each” should be written with small caps

Line 136: You should indicate the manufaturer beside the country origin from.

Line 143: You should indicate the number of biological replicates used in LC-MS experiment.

Line 145: Behind Excel you should indicate corporation and country of origin.

Line 146: The same is for SPSS as I wrote in previous comment.

Results:

Lines 156-157: You should indicate that Tables are Supplementary material (Table S1; Table S2).

Lines 210-212: Description of the Figure 2 has to be improved for the better understanding.

Lines 236-237: Indicate in which Figure.

Lines 254-255: Describe the meaning of the abbreviation CK18, B18, CK48 and etc.

Discussion:

Lines 278-280: Which studies? You have to use appropriate references for this statement. If you thought about your study, please start the sentence with “In our study”.

Lines 278-305: The discussion is poorly written and needs to be expanded. The discussion often restates the obtained results. You should thoroughly discuss your results with those obtained by other authors. Furthermore, biological roles of the metabolites listed in the Results section should be discussed in more detail in this section. Furthermore, this section should be expanded on how this may affect fungal virulence and the host immune response.

Lines 306-310: This should be written as a Conclusion section with included limitations of the present study.

The manuscript submitted for publication in the Journal of Fungi is methodologically well designed and thematically relevant, with the potential to contribute to the understanding of how the rate of Beauveria bassiana appressorium formation affects pathogenicity and cuticular metabolism in two insect hosts. The particular value of the work is reflected in the combination of experimental regulation of appressoria via sulforaphane and non-targeted metabolomics (LC-MS), getting new insights into fungal-insect interactions. The work is relevant, and methodologically good, and addresses a key knowledge gap in the biocontrol and fungal pathogenesis field. However, the manuscript has several shortcomings that need to be addressed. Major revision of the paper before publishing would be beneficial. Language and text style (adequate spacing between the words) should be improved.

The paper can be considered for acceptance after major revision.

Below are my specific comments and questions about the manuscript:

Abstract:

Lines 12-13: Please indicate in abstract that sulforaphane was used as a chemical regulator of appressorium formation.

Introduction:

Lines 53-60: These sentences describe the materials and methods used in this experiment, which do not need to be described in the Introduction section. I recommend stating the hypothesis and knowledge gaps, as well as concluding this section with the aims of the study.

Materials and methods:

Line 67: Please indicate the origin of B. mori larvae.

Line 72-73: To make the manuscript easier to read and follow, an abbreviated group name should be formed for each concentration used.

Line 91: In theses sentence you wrote “Larvae immersed in sterile water were used as the control ck1”. I suggest you to write behind the word control (CK1 group).

Line 94: The same as in previous comment.

Line 102: Avoid starting the sentence with number.

Line 105: Here you wrote 1 minute, at other places you wrote s for seconds and d for days. Decide weather you will use abbreviation or full words throughout the hole manuscript.

Line 117: “Each” should be written with small caps

Line 136: You should indicate the manufaturer beside the country origin from.

Line 143: You should indicate the number of biological replicates used in LC-MS experiment.

Line 145: Behind Excel you should indicate corporation and country of origin.

Line 146: The same is for SPSS as I wrote in previous comment.

Results:

Lines 156-157: You should indicate that Tables are Supplementary material (Table S1; Table S2).

Lines 210-212: Description of the Figure 2 has to be improved for the better understanding.

Lines 236-237: Indicate in which Figure.

Lines 254-255: Describe the meaning of the abbreviation CK18, B18, CK48 and etc.

Discussion:

Lines 278-280: Which studies? You have to use appropriate references for this statement. If you thought about your study, please start the sentence with “In our study”.

Lines 278-305: The discussion is poorly written and needs to be expanded. The discussion often restates the obtained results. You should thoroughly discuss your results with those obtained by other authors. Furthermore, biological roles of the metabolites listed in the Results section should be discussed in more detail in this section. Furthermore, this section should be expanded on how this may affect fungal virulence and the host immune response.

Lines 306-310: This should be written as a Conclusion section with included limitations of the present study.

Reviewer 2 Report

Review

Title: Analysis of the effects of Beauveria bassiana appressorium formation on insect cuticle metabolism based on LC-MS.

The paper evaluated the correlation between appressorium formation and fungal pathogenicity, and explored its impact on insect cuticular metabolism. The paper is relatively well written, presents important aspects about Beauveria bassiana effects on insects through appressorium formation.

The introduction is clear and do not contains unnecessary presentations of previous researches.

The methods are also clear, please explain clear the number of test insects used.

What was the reason of using the presented concentrations? Do they had previously used, or they are recommended for practical use?

Why 5 test worms were inoculated with bacteria for each treatment? Please explain in more details.

What immersion method means, do you have citations here?

At sample processing, 10 larvae were again used. It is confusing again the sample size of each treatment. Please clarify.

The data analyses again needs clarifications. Where the data normally distributed, why nor Anova or Kruscal-Wallis test are selected for data comparison? Why not an r was used instead of SPSS?

What is the reason to use the PCA?

Please increase figure sizes, and axis labeling. Please complete figure legends to be self-explanatory.

I cannot see PCA figures? I am using PCA, but the present forms are not familiar for me.

The Discussion part is too small, I think needs much more details to compare your results with others.

Review

Title: Analysis of the effects of Beauveria bassiana appressorium formation on insect cuticle metabolism based on LC-MS.

The paper evaluated the correlation between appressorium formation and fungal pathogenicity, and explored its impact on insect cuticular metabolism. The paper is relatively well written, presents important aspects about Beauveria bassiana effects on insects through appressorium formation.

The introduction is clear and do not contains unnecessary presentations of previous researches.

The methods are also clear, please explain clear the number of test insects used.

What was the reason of using the presented concentrations? Do they had previously used, or they are recommended for practical use?

Why 5 test worms were inoculated with bacteria for each treatment? Please explain in more details.

What immersion method means, do you have citations here?

At sample processing, 10 larvae were again used. It is confusing again the sample size of each treatment. Please clarify.

The data analyses again needs clarifications. Where the data normally distributed, why nor Anova or Kruscal-Wallis test are selected for data comparison? Why not an r was used instead of SPSS?

What is the reason to use the PCA?

Please increase figure sizes, and axis labeling. Please complete figure legends to be self-explanatory.

I cannot see PCA figures? I am using PCA, but the present forms are not familiar for me.

The Discussion part is too small, I think needs much more details to compare your results with others.

Round 2

Reviewer 1 Report

I have no suggestions.

I have no comments

Reviewer 2 Report

Thank you for your corrections.

I consider, that the authors made significant changes, and considered all my recommendations. I appreciated they effort and answer to my questions.

The paper is much clear and all the figures, including they explanations were corrected.

I do not have any further comments.

Thank you for your corrections.

I consider, that the authors made significant changes, and considered all my recommendations. I appreciated they effort and answer to my questions.

The paper is much clear and all the figures, including they explanations were corrected.

I do not have any further comments.